# Cognitive Mechanisms in Entrepreneurship Competence: Its Implication for Open Innovation

José Alves [1] and Wenjing Yang [1,2,*]

1    Faculty of Business, City University of Macau, Macau 999078, China; b18092100272@cityu.mo
2    East China Economic Management Journal Editorial Office, Hefei 230022, China
*    Correspondence: shuttlediplomacy@163.com; Tel.: +(853)-6562-9115

**Abstract:** From the perspective of cognitive mechanisms, this research discusses the effects of cognitive flexibility, entrepreneurial self-efficacy, and optimism on entrepreneurship competence. Qualitative interviews have revealed the general relationship between key concepts. SEM analysis further shows that cognitive flexibility has a positive direct effect on entrepreneurial self-efficacy; entrepreneurial self-efficacy has a positive direct effect on entrepreneurship competence; cognitive flexibility has a positive direct effect on entrepreneurship competence; cognitive flexibility has a positive indirect effect on entrepreneurship competence through entrepreneurial self-efficacy; optimism moderates the relationship between cognitive flexibility and entrepreneurship competence, with an inverted U-shaped moderation effect. According to the results, this research offers some suggestions to improve entrepreneurs' entrepreneurship competence from the perspective of cognition and positive psychology, hoping that through the overall improvement of entrepreneurs' entrepreneurship competence, it can be beneficial to the open innovation atmosphere of the whole society.

**Keywords:** entrepreneurship competence; cognitive flexibility; entrepreneurial self-efficacy; optimism; open innovation

## 1. Introduction

Entrepreneurship is an activity aimed at creating economic growth; it can improve productivity and create employment opportunities [1]. Entrepreneurship is one of the dynamics of social open innovation [2–4]. Although entrepreneurship has certain economic and social benefits, the current situation of entrepreneurship in practice is not optimistic [5].

According to the data published by *Fortune*, the failure percentage of entrepreneurship worldwide is 70% [6]. CB Insights has summarized the failure conditions of 101 start-ups: nearly half of the many reasons for the failure of start-ups are directly related to the entrepreneurship competence of their founders. For example, the use of contacts and networks is relatively scarce, the opinions of other managers or employees are rarely listened to, the needs of customers are ignored, the enthusiasm for entrepreneurial activities is lacking, and the ability to command the overall situation during the procedure of operation as well as management tends to be low [6].

The lack of entrepreneurship competence of entrepreneurs is mainly reflected in two aspects, which are embodied in the two stages of entrepreneurial enterprise growth. The first stage is the establishment phase of the enterprise. During such a stage, entrepreneurs need to focus on entrepreneurial opportunities, make full use of social network relations, mobilize social capital, make use of surrounding available resources, and identify and develop entrepreneurial opportunities. Most entrepreneurs can find and identify entrepreneurial opportunities at this stage, but it is difficult to overcome the constraints of insufficient innovative resources and effectively organize entrepreneurial activities. The second stage is the growth stage of the enterprise. After the start-up enterprises enter the growth stage, many entrepreneurs are still limited to the thinking mode of the enterprise in the initial stage

and ignore the internal organization and management of the enterprise. Entrepreneurs' shortage of ability during the two phases of the entrepreneurial procedure will eventually lead to entrepreneurial failure [6].

In view of this, to improve entrepreneurial success rates and thus increase the effects of entrepreneurship on economic development and social progress as well as open innovation [7–11], an important starting point is the improvement of the entrepreneurship competence of entrepreneurs.

If we are considering how to improve entrepreneurs' entrepreneurship competence, it tends to be necessary to explore the action paths of these factors on entrepreneurship competence, starting from the antecedent variables and influencing factors of entrepreneurship competence. Only by clarifying this mechanism can we determine the means of the gradual development from possible root causes and preconditions to the improvement of entrepreneurship competence. Nevertheless, via a literature search, it has been found that there are few discussions on the antecedent variables and influencing factors of entrepreneurship competence [12].

Successful business cases show that entrepreneurs with excellent entrepreneurial abilities have obvious traits in common: they do not stick to inherent concepts, they recognize that there are a variety of options and possibilities, and they are willing to embrace open innovation [13]; that is, they have strong cognitive flexibility [14]. Through flexible cognition, they show excellent entrepreneurial ability while controlling the developmental direction of the enterprise. On the other hand, in the theoretical field, entrepreneurial research focuses on the cognitive attributes of entrepreneurs and seeks to summarize the cognitive features of entrepreneurs using norms and rationality [15]. Researchers believe that cognition plays a vital role during the evolution of a specific ability [16]. Cognitive flexibility is regarded as a significant variable within the field of entrepreneurial cognition [15]. Cognitive traits related to cognitive flexibility affect the development of entrepreneurial activities [15].

At the same time, cognitive flexibility is also a prerequisite for entrepreneurial self-efficacy [17]. Cognitive flexibility tends to have a positive predictive function on entrepreneurial self-efficacy [17]. Entrepreneurial self-efficacy promotes the generation of entrepreneurial capability [18]. Meanwhile, typical successful cases of entrepreneurial practice show that entrepreneurs with excellent entrepreneurship competence have a great sense of self-efficacy [19].

It is worth mentioning that, in the current context of widespread entrepreneurial failure, interviews with entrepreneurs have found that 40% of entrepreneurs are worried about entrepreneurial failure [20], indicating that a considerable number of entrepreneurs lack confidence in entrepreneurial success. Nowadays, facing the heavy pressure of the COVID-19 epidemic [21], some individuals who had originally planned to start a business are retreating. They are not fully prepared to start a business given the ongoing crisis, and the long-term restrictive impact of the epidemic cannot be accurately estimated. Facing the heavy pressure of market competition, financing, and insufficient support for all aspects, some individuals have lost most of their original entrepreneurial enthusiasm and lack optimism about entrepreneurship [20]. Theoretical studies show that optimism, as a positive emotion and a psychological characteristic of entrepreneurs, will influence cognitive processes and entrepreneurial success by acting on an individual's entrepreneurial ability [22]. Hence, in the context of COVID-19, optimism is a factor that cannot be ignored in the study of the entrepreneurship competence.

In view of this, this research aims to explore the antecedent variables and influencing elements of entrepreneurship competence and study the action paths of these factors on entrepreneurship competence from a cognitive perspective, so as to find potential areas of focus to improve entrepreneurship competence.

The contribution of this research is mainly reflected in two aspects: research method and the introduction of interdisciplinary concepts. On the one hand, this research adopts a combination of qualitative research and quantitative research. Logically, the two methods

complement each other. For example, the interaction between concepts that qualitative research fails to explore has been further realized through quantitative research, and the abstraction of quantitative research reasoning has been specifically demonstrated in the interpretation of qualitative materials; in the field of function, the research results of the two methods have different audiences. For example, entrepreneurs may feel that the materials of qualitative research are more practical, whereas scholars in the field of entrepreneurship research prefer rigorous data. On the other hand, cognitive flexibility, self-efficacy, and optimism are all concepts in the field of psychological research. Through a literature review, it is found that they rarely appear in the field of entrepreneurship research. This research introduces the concept of psychology into entrepreneurship study and integrates these factors inclined to a positive perspective into a framework for research. It is an innovative experience of positive psychology in the field of entrepreneurship. From the perspective of other disciplines, it may help us to have a more comprehensive understanding of our own disciplines.

The rest of the paper is organized as follows: Section 2, key concepts and research hypotheses; Section 3, method; Section 4, results; and Section 5, discussion.

## 2. Key Concepts and Research Hypotheses

### 2.1. Key Concepts

#### 2.1.1. Entrepreneurship Competence

Bird [23] concludes that competence represents the benchmark of planning and entrepreneurship and represents the high standard of achieving sustainable development and growth. Entrepreneurship competence is also defined as a set of specific attributes related to entrepreneurial success behavior, which is a comprehensive ability. Such as appropriate attitude and motivation, a series of relevant entrepreneurial knowledge and skills to promote the sustained success of entrepreneurs; these attributes are finally characterized by specific entrepreneurial behavior, which will be affected by personal background such as age, education level, and working years, and can be changed through learning, education, and training [24].

#### 2.1.2. Cognitive Flexibility

Spiro [25] believes that cognitive flexibility is likely to be a capability of individuals to reconstruct knowledge and organizational resources in a variety of ways (such as facing complex situations) under the needs of different backgrounds. Later, scholars propose two different types of cognitive flexibility: reactive flexibility and spontaneous flexibility. According to them, reactive flexibility suggests transforming one's way of thinking to adapt to changes in the environment; spontaneous flexibility involves two sets of reactions that can be produced without external clues or missing [26]. Martin and Rubin [27] extend the study of cognitive flexibility from the field of education and teaching to other fields and put forward the concept of cognitive flexibility from a new perspective. They believe that cognitive flexibility is that people have more choices in the actions they should take when dealing with a certain situation; that is, they are willing to adapt to the change of environment and believe that they have the ability to deal with new situations.

#### 2.1.3. Entrepreneurial Self-Efficacy

Boyds and Vozikis [28] define entrepreneurial self-efficacy as the belief intensity that individual believe that he or she could completely play various entrepreneurial roles and accomplish different entrepreneurship assignments. The concept of entrepreneurial self-efficacy, like the implication of self-efficacy, does not indicate a certain personality trait or entrepreneurship behavior capabilities themselves, but tends to refer to the confidences or beliefs in one's own capability formed by synthesizing all kinds of information and on the basis of judgments and evaluations of one's own entrepreneurial behavior competence [29].

2.1.4. Optimism

Scheier and Carver [30] point out that optimism is a relatively unchanging nature trait of people. This trait always tends to make good things happen more easily than bad things. In the face of setbacks and difficulties, individuals will constantly adjust themselves to achieve their goals as much as possible; they put forward the concept of temperament optimism. Other scholars [31] argue that optimism is not a universal personality trait. It is a different tendency of individuals in attribution analysis after events. It is an explanatory style. They define optimism as the stable tendency of individuals in attribution tracing of success or failure.

*2.2. Research Hypotheses*

2.2.1. Cognitive Flexibility and Entrepreneurial Self-Efficacy

In a group of experiments on the impact of childhood trauma on early adulthood resilience and dynamic regulation mechanisms, researchers found that there is a positive correlation between cognitive flexibility and self-efficacy [32]. In the process of studying the characteristics of cognition and the coping styles of military medical personnel involved in COVID-19 prevention and control, it has been noticed that individuals with higher cognitive flexibility have stronger psychological toughness and creativity, higher self-efficacy, and can more easily access positive coping [33]. During the study of the positive effect of power on cognitive flexibility, researchers have indicated that self-efficacy and cognitive flexibility promote each other [17]. With the improvement of cognitive flexibility, individuals become more confident in dealing with problems; they have faster responses and stronger insights. A study has found that individuals with stronger cognitive flexibility also have a higher sense of efficacy as regards their own creative performance. Similarly, individuals with a better sense of innovation efficacy have a higher level of cognitive flexibility [34]. Entrepreneurship itself is a creative activity, so cognitive flexibility may have a positive predictive effect on entrepreneurial efficacy. When individuals evaluate their abilities, they often rely on their own emotions or cognition [35]. Positive emotions or flexible cognition can enhance entrepreneurial self-efficacy, and vice versa [18]. Cognitive flexibility tends to be a prerequisite of efficacy belief, which is characterized by selectivity and controllability [36]. After mastering the corresponding basic knowledge, individuals with cognitive flexibility can capitalize on their own unique advantages, enhance their confidence, cope with uncertain factors in the entrepreneurial process, adapt to the diversity of knowledge and information, and improve their entrepreneurial self-efficacy. Cognitive flexibility can also help to reduce anxiety and other negative emotions, and help individuals build confidence, which can enhance positive emotions and entrepreneurial self-efficacy [37].

Accordingly, research hypothesis 1 is put forward.

**Hypothesis 1 (H1).** *Cognitive flexibility has a positive direct effect on entrepreneurial self-efficacy.*

2.2.2. Entrepreneurial Self-Efficacy and Entrepreneurship Competence

Self-efficacy plays an important role in promoting entrepreneurial awareness and entrepreneurial traits in the context of entrepreneurial competence [38]. Self-efficacy helps entrepreneurs with making full use of the external environment and resources, with the purpose of achieving entrepreneurial goals, handling the relationship between performance management and emotional management, and enhancing their ability to resist external risks [39]. Self-efficacy determines the effort and persistence of entrepreneurs in the face of entrepreneurial difficulties [40]. Only when entrepreneurs accurately recognize their own abilities and have firm confidence in their entrepreneurial success can they stand out in the entrepreneurial "army" [38]. Innovation is the fuel of entrepreneurship, and a good sense of creativity and self-efficacy is the core of individual innovation [41]. Self-efficacy helps individuals learn innovation and entrepreneurship skills more actively by improving their

self-awareness and self-evaluation [39]. Self-efficacy can stimulate individuals' enthusiasm for innovation and entrepreneurship, guiding them towards active practice and promoting their innovation and entrepreneurship ability [41]. As a kind of psychological capital, self-efficacy tends to have an important positive effect on entrepreneurship ability. Self-efficacy can encourage individuals to actively coordinate interpersonal relationships and fully and effectively express themselves, which is an important exogenous variable in promoting entrepreneurial ability [40]. Self-efficacy promotes entrepreneurial ability by changing entrepreneurs' cognition. In a group of entrepreneurship studies on college students, it was found that, as regards students' entrepreneurship practice, self-efficacy can significantly affect college students' entrepreneurial intention and ability [42]. Entrepreneurial self-efficacy may play an important role in identifying entrepreneurial opportunities. Empirical research shows that entrepreneurial self-efficacy can positively predict entrepreneurial opportunity identification, which is an essential part of entrepreneurial ability [43].

Hence, research hypothesis 2 is put forward.

**Hypothesis 2 (H2).** *Entrepreneurial self-efficacy has a positive direct effect on entrepreneurship competence.*

### 2.2.3. Cognitive Flexibility and Entrepreneurship Competence

Cognitive flexibility is likely to be a vital psychological trait to individuals' ability to overcome uncertainty and successfully complete unstructured complex tasks [37]. It tends to play a significant role in the formation of individual entrepreneurial ability, as well as in the coordination of environmental adaptation [37]. Individuals with high cognitive flexibility have high creativity [32]. Cognitive flexibility emphasizes that individuals can think about problems from different point of view, put forward a variety of solutions to problems, avoid core rigidity, and enhance the adaptability of start-ups [37]. In research on the relationship between cognitive flexibility and ability, researchers have maintained that high individual cognitive flexibility promotes the strategic adaptability of start-ups, helping them adapt to the external environment and respond to the dynamic competitive strategy of competitors, while playing a lead role in the change of environment, creating an environment suitable for enterprise development, and improving the adaptability of enterprises. It also tends to have an essential impact on the evolution and allocation of dynamic capabilities [37]. The influence of cognitive flexibility on ability is mainly reflected in environmental adaptability [44]. It has been pointed out that cognitive flexibility, as a kind of ability, allows one to adapt to the transformations of the external environment as regards goal orientation, which is the dynamic strategic view of an enterprise, emphasizing not only the ability to respond to the environment and develop insight, but also the ability to develop and implement a strategy [45]. Cognitive flexibility consists of flexible consciousness, flexible will, and self-efficacy [33]. These elements suggest that individuals with higher cognitive flexibility have a stronger sense of contact with the external dynamic environment. The micro-composition of cognitive flexibility determines the enhancement of cognitive flexibility in response to the environment. Cognitive flexibility gives individuals the ability to develop multiple knowledge attributes. Individuals with higher cognitive flexibility maintain a diversity of perceptions, data, rule sets, as well as behavior patterns in their cognitive reserves, and they are better at making new solutions to deal with tough questions and non-determinacy [32]. Cognitive flexibility tends to be the basis for entrepreneurs to interpret and form strategic responses. A strong level of cognitive flexibility helps entrepreneurs to take more diverse approaches to recognizing and interpreting relevant information and promotes their free conversion in different thinking and searching modes, enabling them to properly handle the trade-off between exploratory activities and mining activities, and then produce novel solutions [37]. Entrepreneurs with high levels of cognitive flexibility have the ability of entrepreneurial bricolage [46]. Entrepreneurs with strong cognitive flexibility can better reflect on existing strategic actions and absorb cues

from the environment, which is positively related to their ability to undertake dynamic task feedback.

Accordingly, research hypothesis 3 is put forward.

**Hypothesis 3 (H3).** *Cognitive flexibility has a positive direct effect on entrepreneurship competence.*

### 2.2.4. The Mediating Role of Entrepreneurial Self-Efficacy

Entrepreneurial self-efficacy is not only a predictive variable, but also has an effect on entrepreneurial decision-making and performance; it is also an outcome variable, which is affected by gender role identification, cognitive style, risk-taking tendency, and other factors [47]. It can be seen that entrepreneurial self-efficacy plays an intermediary role in entrepreneurship research. The mediating effect of entrepreneurial self-efficacy has been explored in many studies. Entrepreneurial self-efficacy plays an intermediary role in the relationship between the social network and individuals' entrepreneurial intention [48]. Entrepreneurial self-efficacy plays a vital intermediary role between entrepreneurial experience and effect reasoning, as well as between functional experience and effect reasoning [49]. Entrepreneurial self-efficacy plays an intermediary role in the individual's ability to navigate between the service environment and their own entrepreneurial orientation [50]. Entrepreneurial self-efficacy plays a mediating role in the impact of entrepreneurial learning on entrepreneurial orientation [51]. Individual entrepreneurial experience has a positive impact on the development of entrepreneurial opportunities through entrepreneurial self-efficacy [49]. Entrepreneurial self-efficacy plays an intermediary role between social support and entrepreneurial persistence [52]. Entrepreneurial passion further affects the performance of start-ups by mediating entrepreneurial self-efficacy [52].

In this study, based on the analysis of hypothesis 1 and hypothesis 2, the cognitive flexibility of entrepreneurs has been found to improve entrepreneurial self-efficacy, and individuals with high entrepreneurial self-efficacy tend to have higher entrepreneurship competence. Cognitive flexibility influences entrepreneurship competence by influencing self-efficacy.

Accordingly, research hypothesis 4 is put forward.

**Hypothesis 4 (H4).** *Cognitive flexibility has a positive indirect effect on entrepreneurship competence through entrepreneurial self-efficacy.*

### 2.2.5. The Moderating Effect of Optimism

Optimism is often introduced in the research on psychology, and its moderating effect is common. In entrepreneurship research, the regulatory role of optimism has also been mentioned. Entrepreneur optimism plays a significant negative regulatory role between the learning network and entrepreneurial ability [53]. Moderate optimism has a positive regulatory effect on the relationship between entrepreneurial action learning and family entrepreneurial ability; excessive optimism has a negative regulatory effect on the relationship between entrepreneurial action learning and family entrepreneurial ability [54]. Team leaders' optimism positively regulates the relationship between industrial and functional experience heterogeneity and new enterprise performance [55].

In research, optimism has been found to have a direct effect on cognitive flexibility [56]; that is, optimism can positively predict cognitive flexibility. Optimistic entrepreneurs have high cognitive flexibility, actively carry out entrepreneurial patchwork, have increased access to entrepreneurial information and resources, and adhere to their entrepreneurial beliefs, all of which are closely related to the entrepreneurial abilities of entrepreneurs [57]. Entrepreneurs who are driven by moderate optimism can constantly generate their own internal motivation, which is conducive to action, and are more inclined to look register and respond to the environment around them with an appreciative eye and attitude, so as to broaden the scope of their cognition and improve their entrepreneurial ability [57].

Excessive optimism will lead to more cognitive bias, and unrealistic expectations of success, which will lead to the upgrading of entrepreneurs' commitment [57].

Hence, research hypothesis 5 is put forward.

**Hypothesis 5 (H5).** *Optimism moderates the relationship between cognitive flexibility and entrepreneurship competence with an inverted U-shaped moderation effect.*

Based on the five hypotheses, the theoretical model of this research has been constructed (Figure 1).

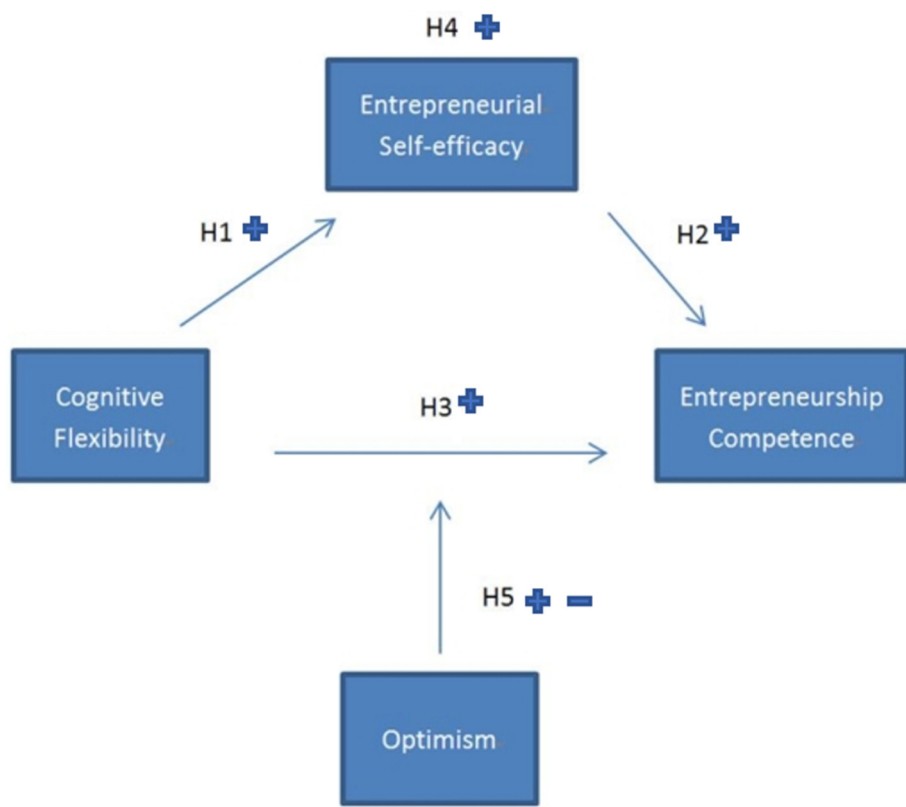

**Figure 1.** Theoretical model.

### 3. Method

This research relies on a hybrid approach: a combination of qualitative and quantitative methods. Both the qualitative method and the quantitative method have defects, and the merits of one will help improve the other [58]. This research preliminarily analyzes the relationship between concepts through qualitative interviews and gives the audience a vivid understanding of abstract concepts through the expressions of the respondents. Then, quantitative research further verifies the relationships between variables; that is, it employs statistical data to effectively verify whether the hypothesis is supported or not. The hybrid approach means this research not only has the accuracy and elegance of quantitative research, but it also maintains the liveliness of qualitative research. Researchers prefer rigorous data, whereas entrepreneurs benefit more from lived experience.

#### 3.1. Qualitative Method

In the qualitative research section, we adopt the interview method.

We select 20 entrepreneurs (Supplementary Materials: Basic Information of Interviewees) of different genders, ages, and education levels, from different industries and locations, from a list of more than 100 candidates introduced via an organization of en-

trepreneurs. Most of the entrepreneurs are from mainland China; another entrepreneur is from Macau, China, and a female entrepreneur of Chinese nationality is from Cardiff, UK.

We guide the interviewees to comment on certain themes via targeted questions. The interview outline is as follows: How do you feel about your cognitive flexibility? Perhaps this concept is too abstract. For example, are you a flexible person in daily life? Does your mind turn fast? Are you stubborn? Do you have many ideas when you encounter problems? When in trouble, do you think you can control the situation? Do people you know think you are a flexible person? What is your confidence in starting your own business? Do you feel that you are fully prepared to start businesses? Do you think you can succeed? In daily life, do you think you are an optimist or a pessimist? Do acquaintances describe you as optimistic? Do you think optimism or pessimism has any impact on your entrepreneurship? Talk about your entrepreneurial process, i.e., how to find business opportunities, how to finance, and business management details. Overall, how do you rate your entrepreneurial ability? Do your current entrepreneurial achievements show your ability in entrepreneurship?

We adopted a content analysis method to analyze the interview materials. Content analysis is chiefly conducted by researchers through reading, listening, and observing, and then, on the basis of subjective feelings, understanding, experiences and analyses, they interpret, evaluate, and explore the substantive content within the information [59]. We tried to identify some expressions from the respondents that captured the essential meaning of each concept, and we sorted the relationships between concepts through the clues provided by the interviewees.

### 3.2. Quantitative Method

In the quantitative research part, we adopted the method of a questionnaire survey and data analysis.

The main research tool was a questionnaire. We composed the questionnaire on the basis of four variables; the questionnaire also contained some items related to population information. The measurement of variables depended on the following: the independent variable of cognitive flexibility, which was measured by the cognitive flexibility questionnaire constructed by Dennis and Wal [36], who divided the measurement of cognitive flexibility into two dimensions (alternatives and control); the dependent variable of entrepreneurship competence, which was measured by the scale developed by Xie and Wang [54], who divided entrepreneurship competence into five dimensions (opportunity, commitment, conception, financing, and operation); the mediating variable of entrepreneurial self-efficacy, which was measured by Liñán and other scholars [47]; and the moderating variable of optimism, which was measured by the life orientation scale developed by Scheier, Carver, and Bridges [60]. The questionnaire appears in Appendix A for reader reference.

The sampling methods used were convenience sampling and snowball sampling in the form of non-probability sampling. Causal analysis and exploratory surveys do not have high requirements as regards the representativeness of samples. Their main task is not to estimate the overall indicators, but to form a certain point of view, which is suitable for convenience sampling [61]. Two main methods of obtaining samples were used in this research. One was to distribute 300 questionnaires to entrepreneurs and their friends (who were also entrepreneurs) with the help of an MBA institution. The other was to participate in an entrepreneur conference, at which 200 paper questionnaires are distributed. Participants were spread all over China, including several entrepreneurs who originally come from Pakistan and the UK. Overall, this research has distributed 500 questionnaires; 356 questionnaires were retrieved, and 33 invalid questionnaires eliminated, which left 323 valid questionnaires (Supplementary Materials: General Situations of the Sample). The effective recovery rate was 64.6%. The main criteria for excluding questionnaires were: the questionnaire was not completed; and the questionnaire options were basically the same, indicating that the attitude is obviously perfunctory.

SEM and Amos software are employed for data analysis and hypothesis testing.

## 4. Results

### *4.1. Qualitative Result*

Based on our questions, combined with the interview outline, the interviewees gave detailed statements (Supplementary Materials: Collection of Interview Materials). According to the research requirements, we extracted and classified the key statements in order to capture an effective concept description and the relationship between concepts. It should be noted that, since this is qualitative research, we could determine the relationships between concepts (for example, concept A being present to a higher degree and concept B being present to a higher degree are reflected in the respondents' statements, whereas concept A to a lower degree and concept B to a lower degree coexist, so we infer that the two concepts of A and B may be positively correlated); from our research, we can infer that one concept affects or determines another. Our classification and evaluation of the respondents' statements were also based on our subjective and general judgments, based on our deep scrutiny of the previous literature.

### 4.1.1. Relationship between Cognitive Flexibility and Entrepreneurial Self-Efficacy

Through a simple frequency analysis, we found that 7 of the 20 interviewees indicated a relationship between cognitive flexibility and entrepreneurial self-efficacy. We selected the typical expressions of three of these for concept classification and evaluation, as shown in Table 1.

The evaluation results indicate that individuals with higher cognitive flexibility have higher entrepreneurial self-efficacy; individuals with low cognitive flexibility have low entrepreneurial self-efficacy. Based on this, we tend to believe that there is a positive correlation between cognitive flexibility and entrepreneurial self-efficacy.

**Table 1.** Interview material distilling A.

| Material Code | Entrepreneur Code | Expression | Classification | Evaluation |
|---|---|---|---|---|
| Material 1 | Entrepreneur 8 | I thought I should have a better choice. | cognitive flexibility (alternatives) | The entrepreneur's expression reflects good cognitive flexibility and high entrepreneurial self-efficacy. |
| | | Instead of sticking to the tradition and waiting to die, it is better to find another way out and find a better development industry. | cognitive flexibility (control) | |
| | | I thought if I embrace this dream, I would succeed. | self-efficacy | |
| | | I received a good education abroad and obtained a master's degree, which suggests my learning ability is beyond doubt. | self-efficacy | |
| | | But I believed that after a period of study, I would certainly be able to master the development law of this industry and do a good job in this company. | entrepreneurial self-efficacy | |

**Table 1.** *Cont.*

| Material Code | Entrepreneur Code | Expression | Classification | Evaluation |
|---|---|---|---|---|
| Material 2 | Entrepreneur 9 | Yes, people in our village say my brain is alive. | cognitive flexibility | The entrepreneur's expression reflects good cognitive flexibility and high entrepreneurial self-efficacy. |
| | | We peasants always say "the living can't be suffocated by urine". | cognitive flexibility (control) | |
| | | I had no way, but the bank had a way. | cognitive flexibility (alternatives) | |
| | | I patted my chest and told my uncle that I would be able to run the factory. Others are successful in business. Why can't I? My brain is no worse than others, and I am hard-working. If a clever man is willing to insist and bear hardships, will there be anything he can't do? | entrepreneurial self-efficacy | |
| Material 3 | Entrepreneur 14 | It was like playing a card game, and then I was judged to be the group with low cognitive flexibility. | cognitive flexibility | The entrepreneur has participated in psychological experiments, and the results show that his cognitive flexibility is low; the entrepreneur's cognitive flexibility in self-assessment is low; the entrepreneur's expression of cognitive flexibility tends to use negative expressions. The entrepreneur's entrepreneurial self-efficacy is low, and the whole expression is uninspired. |
| | | You can also say that I am the same outside and inside. I don't look like a smart man with all sides, do I? | cognitive flexibility | |
| | | This B & B is not my idea. | cognitive flexibility | |
| | | I didn't know what this building should be used for. | cognitive flexibility | |
| | | But it seems that except for transforming it into a B & B, I can't think of a better way to develop its value. | cognitive flexibility | |
| | | But from the beginning, I felt that the possibility of success of this project was not high. | entrepreneurial self-efficacy | |
| | | Knowing that the investment will not succeed, | entrepreneurial self-efficacy | |

### 4.1.2. Relationship between Entrepreneurial Self-Efficacy and Entrepreneurship Competence

Through simple frequency analysis, we found that 14 of the 20 interviewees indicated a relationship between entrepreneurial self-efficacy and entrepreneurship competence. We selected typical expressions of five of these for concept classification and evaluation, as listed in Table 2.

The evaluation results suggest that individuals with higher entrepreneurial self-efficacy have stronger entrepreneurship competence; individuals with low entrepreneurial self-efficacy have low entrepreneurship competence. Based on this, we deduce that there is a positive correlation between entrepreneurial self-efficacy and entrepreneurship competence.

**Table 2.** Interview material distilling B.

| Material Code | Entrepreneur Code | Expression | Classification | Evaluation |
|---|---|---|---|---|
| Material 4 | Entrepreneur 2 | In most cases, I am a confident person. | self-efficacy | The entrepreneur shows high self-efficacy and excellent entrepreneurship competence. |
| | | The continuous superposition of successful experiences has strengthened my self-confidence. | self-efficacy | |
| | | So I also have strong self-confidence in entrepreneurship. | entrepreneurial self-efficacy | |
| | | I thought to myself, why not use the huge market formed by these student resources to set up my own English training institution? | entrepreneurship competence (opportunity ability, conception ability) | |
| | | I can do well in other people's training institutions, so I can operate my own training institutions well. | entrepreneurial self-efficacy | |
| | | In the process of starting the organization, I found a very key chance. The epidemic has created the rise of online classes. Instead of renting visible teaching venues at a high cost, why not set up an online training institution? | entrepreneurship competence (opportunity ability, conception ability) | |
| | | So I gathered excellent teachers in the industry and set up the current English school. | entrepreneurship competence (operation ability) | |
| | | With our concerted efforts, the school is running very well. It's much better than when I was a teacher in someone else's training institution. | entrepreneurship competence | |
| | | I also want to further expand the scale of the school and turn our training institution into a general subject school covering all subjects in primary and secondary schools. | entrepreneurship competence (conception ability) | |
| Material 5 | Entrepreneur 5 | If I can open a Vietnamese restaurant, it will undoubtedly satisfy my stomach and increase my family's income. Why not? | entrepreneurship competence | The entrepreneur shows good self-efficacy and good entrepreneurship competence. |

**Table 2.** *Cont.*

| Material Code | Entrepreneur Code | Expression | Classification | Evaluation |
|---|---|---|---|---|
| | | I told him it wasn't difficult. The college entrance examination was so difficult that I still have been admitted to Peking University. This entrepreneurship was nothing more than an examination. As long as I work hard, I will be able to do well. | entrepreneurial self-efficacy | |
| | | It takes money to open a restaurant. I used my good popularity among my colleagues and fellow villagers to raise a million. Opening a restaurant requires a good location. I repeatedly persuaded a local colleague in Shanghai who has a shop on Nanjing Road to rent the shop to me at a low price. | entrepreneurship competence (financing ability) | |
| | | The most important thing to open a restaurant is a good cook. I did my best to summon my mother and aunt from Guangxi to Shanghai to become the chefs of our restaurant. | entrepreneurship competence (operation ability) | |
| | | Facts have proved that my judgment is correct. Our Vietnamese restaurant is booming. | entrepreneurship competence | |
| Material 6 | Entrepreneur 11 | I am engaged in agricultural science and technology. As a part-time professor in a university, I also teach agriculture. This venture to build an ecological farm is actually an in-depth excavation on the basis of the old career. I also have an agricultural technology company in my hand. In the past 20 years, I have also experienced four or five entrepreneurships, large and small, with successes and failures. Up to now, I am nearly half a hundred years old, and I can say that I have accumulated a lot of experience. | entrepreneurship competence | The entrepreneur is undoubtedly a double model of entrepreneurship theory and practice. He shows high entrepreneurial self-efficacy and entrepreneurship competence. |

**Table 2.** *Cont.*

| Material Code | Entrepreneur Code | Expression | Classification | Evaluation |
| --- | --- | --- | --- | --- |
| | | I'm familiar with entrepreneurship. I know the way. By the way, I am also an entrepreneurship tutor in a university in Singapore. | entrepreneurial self-efficacy | |
| | | Ecological farms can be said to keep up with the trend of the times. You see, ecotourism is so popular now. | entrepreneurship competence (opportunity ability) | |
| | | Our farm business starts with eco-tourism, positioning the high-end tourism market and developing business leisure tourism services. Meanwhile, we use the natural pollution-free ecological environment of the farm to produce high-quality organic food and green food, which can be provided to high-end restaurants and hotels. Of course, we also build amusement parks and deepen the development of tourism services. | entrepreneurship competence (conception ability) | |
| | | And as you know, my doctoral program is management. I have both theory and experience in business management. | entrepreneurial self-efficacy; entrepreneurship competence (operation ability) | |
| | | Our farm tries to jump out of agriculture and grasps agriculture, and transforms traditional agriculture with industrialized mechanisms, means, and elements. We use information technology to effectively manage procurement, production and sales, and vigorously promote standardized production. In terms of human resources, we generally pay attention to staff training, replace managers with operators, and regard people as key resources rather than utilization objects. | entrepreneurship competence (operation ability) | |

**Table 2.** *Cont.*

| Material Code | Entrepreneur Code | Expression | Classification | Evaluation |
|---|---|---|---|---|
| | | Our farm not only has economic value, but also I have built it into an education base. We have signed scientific research practice cooperation agreements with colleges and universities such as Hainan University. | entrepreneurship competence (conception ability) | |
| Material 7 | Entrepreneur 1 | I thought it couldn't be done. It really couldn't be successful. How many successful entrepreneurs have you seen? Not everyone is Ma Yun. We haven't done anything except reading ("reading" means receiving school education—researcher's note) for twenty or thirty years. Two ears don't hear things outside the window, and ten fingers don't touch the spring water. (Chinese proverbs. It means to do nothing but study—researcher's note) If we do farming successfully, where will we put the farmers who do it all their life? Is there any reason? | entrepreneurial self-efficacy | The entrepreneur has low entrepreneurial self-efficacy and lacks entrepreneurship competence. |
| | | The results can be imagined. Without experience, many crabs died. The ones that survived were small and less crab roe. They were no better than big brand hairy crabs such as "Yangcheng Lake". Our hairy crab online shop is also closing down, and no one cares. | entrepreneurship competence | |
| Material 8 | Entrepreneur 14 | The current business situation has verified my initial assumption. It is indeed a wrong decision to convert this building into a B & B. | entrepreneurial self-efficacy; entrepreneurship competence | The entrepreneur has a low entrepreneurial self-efficacy, negative attitude, and low entrepreneurship competence. |
| | | will it be better? I don't think so. | entrepreneurial self-efficacy | |
| | | And you see, this street is full of B & B, homogenization competition is so fierce, and we have no characteristics. Unless other better hotels are full, customers will not choose us. | entrepreneurial self-efficacy; entrepreneurship competence | |

**Table 2.** *Cont.*

| Material Code | Entrepreneur Code | Expression | Classification | Evaluation |
|---|---|---|---|---|
| | | We haven't studied hotel management, and we're clumsy to help at the front desk. | entrepreneurship competence | |
| | | First and last, I was not optimistic at the beginning. I didn't have any motivation and passion. I'm too lazy to think about it again. | entrepreneurial self-efficacy | |

### 4.1.3. Relationship between Cognitive Flexibility and Entrepreneurship Competence

Through simple frequency analysis, we found that 11 of the 20 interviewees suggested a relationship between cognitive flexibility and entrepreneurship competence. We selected typical expressions of four of these for concept classification and evaluation, as listed in Table 3.

Our evaluation results suggest that individuals with higher cognitive flexibility have higher entrepreneurship competence; individuals with lower cognitive flexibility have lower entrepreneurship competence. Based on this, we infer that there is a positive correlation between cognitive flexibility and entrepreneurship competence.

**Table 3.** Interview material distilling C.

| Material Code | Entrepreneur Code | Expression | Classification | Evaluation |
|---|---|---|---|---|
| Material 9 | Entrepreneur 17 | But the stock is always too risky, not a long-term plan. I thought I can't let the money idle and take risks. I can start a business and create more and more stable wealth. | cognitive flexibility (alternatives) | The entrepreneur shows high cognitive flexibility and good entrepreneurship competence. |
| | | We women should take into account children and families. We can't afford too much business. We might as well do some small businesses within our ability. | cognitive flexibility (alternatives) | |
| | | I began to think about what is more conducive to the current economic situation, what people just need, what cannot be solved in online stores and can only be completed in the entity. | entrepreneurship competence (opportunity ability) | |
| | | Aren't these venues and human resources suitable for a postpartum care center? | entrepreneurship competence (conception ability) | |
| | | We invited special teachers from Dalian to train the staff. In addition, I went to a postpartum care center in Shanghai for three months to study. | entrepreneurship competence (operation ability) | |

**Table 3.** *Cont.*

| Material Code | Entrepreneur Code | Expression | Classification | Evaluation |
|---|---|---|---|---|
| | | Now the nine rooms in my confinement center are full, and many expectant mothers have made an appointment six months later. | entrepreneurship competence | |
| | | My business is very complete. | entrepreneurship competence (operation ability, conception ability) | |
| | | Women's entrepreneurship should start from the perspective of women. Women can help women. | entrepreneurship competence (opportunity ability, conception ability) | |
| Material 10 | Entrepreneur 19 | But they only saw the benefits within the system, not the broader market outside the system. | cognitive flexibility (alternatives) | The entrepreneur shows good cognitive flexibility, courage and calmness, and strong entrepreneurship competence. |
| | | Why should I stick to the small world of an institution and keep the fixed monthly salary? It's better to rush. I was almost 40 years old. I can't toss about in a few years. | cognitive flexibility (alternatives) | |
| | | I saw the value of Macau, the most economically developed region, without losing order. Most of the people who come to Macau for entertainment are rich people, who are the largest audience in our art auction industry. | entrepreneurship competence (opportunity ability) | |
| | | It's not easy to open an auction company in Macau. We should take care of all aspects. We can't miss any temple. (Chinese proverbs. It means doing business should keep a good relationship with all competent departments—researcher's note) You see, I have made so many friends from all walks of life. Many of them are business relationship. | entrepreneurship competence (operation ability) | |

**Table 3.** *Cont.*

| Material Code | Entrepreneur Code | Expression | Classification | Evaluation |
|---|---|---|---|---|
| | | The loss is a little big, but doesn't entrepreneurship have ups and downs? Hold on! You 90s generation always like to say that you have to finish walking on your knees. (Internet buzzwords. It means that whatever you choose to do, you have to finish it no matter how difficult it is—researcher's note) | entrepreneurship competence (commitment ability) | |
| | | We Chinese often say "farming in the sunny days and reading in the rainy days". Take advantage of this whole depressed leisure time to improve myself. | cognitive flexibility (alternatives) | |
| | | The management of enterprises needs continuous self-improvement, and I practice it. | entrepreneurship competence | |
| Material 11 | Entrepreneur 20 | You know, it's hard to find a job in Britain. People in their own countries can't find a job. Why do they want us foreigners? I continued to engage in the purchasing industry, set up my own studio, and found two other women who are also staying here with their husbands to do purchasing together. | cognitive flexibility (alternatives) | The entrepreneur shows good cognitive flexibility and good entrepreneurship competence. |
| | | What kind of work is not difficult? If you have difficulties, you can find a way to solve them! If there are problem, I'll break them one by one. | entrepreneurship competence (commitment ability) | |
| | | For homogeneous competition, I work out my own characteristics. | entrepreneurship competence (conception ability, operation ability) | |
| | | In order to eliminate the customer's concern about the authenticity of purchasing, my solution is to play the acquaintance card. | entrepreneurship competence (operation ability) | |
| | | Mine has been nine years. Right? You've heard from Bibi. I won't hide it from you. My monthly income is better than your annual income. | entrepreneurship competence | |

**Table 3.** *Cont.*

| Material Code | Entrepreneur Code | Expression | Classification | Evaluation |
|---|---|---|---|---|
| Material 12 | Entrepreneur 6 | I was born as a soldier. After I retired from the army, I had nothing else to do, no skills, and no entrepreneurial ideas. As I like riding motorcycles, I opened this training school. | cognitive flexibility (alternatives) | The entrepreneur's expression shows obvious stubbornness, stupidity, low cognitive flexibility, and poor entrepreneurship competence. |
| | | These motorcycles are still new. | entrepreneurship competence | |
| | | I didn't think so much. | entrepreneurship competence | |
| | | So I can't lower the training fee. | cognitive flexibility | |
| | | My brother said that the root cause was that my EQ was not very high and I couldn't speak politely. I speak in a blunt manner to learners, have a bad attitude, and they won't introduce me to new customers. | cognitive flexibility | |
| | | What can I do? We are soldiers. The bounden duty of soldiers is to obey orders. How can there be so many colorful ideas? | cognitive flexibility (alternatives) | |
| | | If business doesn't work this summer, I'll sell the motors, close the school, and pay my brother some money firstly. | entrepreneurship competence (commitment ability) | |

### 4.1.4. Some Expressions on Optimism

Through simple frequency analysis, we found that 9 of the 20 interviewees indicate the utility of optimism. Optimism has different effects on entrepreneurs' cognitive flexibility and entrepreneurship competence, and on the relationship between them. We selected typical expressions of four of these, classifying the concepts and evaluating them, as listed in Table 4.

Our evaluation results indicate that, in the relationship between cognitive flexibility, entrepreneurship competence and optimism, optimism plays a positive role in some cases, which is consistent with good cognitive flexibility and excellent entrepreneurship competence; in extreme cases, excessive optimism has a negative effect on entrepreneurs' entrepreneurship competence and on the whole entrepreneurial process. Based on this, we hold that the effect of optimism is not intrinsically positive or negative but varies according to its degree.

**Table 4.** Interview material distilling D.

| Material Code | Entrepreneur Code | Expression | Classification | Evaluation |
|---|---|---|---|---|
| Material 13 | Entrepreneur 7 | If I summarize the outstanding advantages of Inner Mongolia people, I think it is optimism. | optimism | The entrepreneur has obvious optimism, good cognitive flexibility, and excellent entrepreneurship competence. The entrepreneur believes optimism plays a positive role in his entrepreneurship. |
| | | Oh, this thing is too important. I have been herding sheep on the grassland since I was a child. There are eagles in the sky and wolves on the ground. How could I persist if I don't have the spirit of optimist and believe that the heaven will bless me? | optimism | |
| | | Why do you think so much? Just do it! | entrepreneurship competence | |
| | | If you are willing to work hard, the heaven will help you! I believe this. I don't think the poor children on the grassland will have bad luck. How much worse? You hit the bottom and can bounce. | optimism | |
| | | Barefoot people are not afraid of those wearing shoes. (Chinese proverbs. It means that the poor are braver than the rich because the poor have nothing to worry about—researcher's note) What am I afraid of?! | optimism | |
| | | Hope is around the corner, isn't it? | optimism | |
| | | I began to look for new business opportunities. | cognitive flexibility | |
| | | After all, I'm lucky. Without culture and background, I have come to this day with a mentality full of hope. | optimism | |
| Material 14 | Entrepreneur 15 | I admit that the hotel industry is hard to do after the epidemic, but my home stay was opened in the year of the epidemic. In other words, I have thought of all the negative factors you can think of, really, but I insisted on doing it. | cognitive flexibility (alternatives); entrepreneurship competence (commitment ability) | The entrepreneur has certain cognitive flexibility and an indomitable entrepreneurial commitment. The entrepreneur's optimism undoubtedly inspires her entrepreneurial process. |
| | | I am an idealist. I firmly believe that dreams can be helped by God. | optimism | |

**Table 4.** *Cont.*

| Material Code | Entrepreneur Code | Expression | Classification | Evaluation |
|---|---|---|---|---|
| | | Now Guanerjie is really a little famous. Everyone knows that it carries a designer's own dream. Guanerjie also won a prize in a home stay design competition. As for performance, although we can't say it's full every day, we have a stable source of customers. | entrepreneurship competence | |
| | | I believe Guanerjie will be better and better, and I will be better and better. If a person only insists on doing one thing all her life, what do you think is the probability that she can do it well? I think it's 100%. | optimism | |
| Material 15 | Entrepreneur 3 | I put the reason for the closure down to four words, blind optimism. (Chinese four words 盲目樂觀, when translated into English, they are two words—researcher's note) | entrepreneurship competence; over optimism | When the entrepreneur does project post-mortem, she regrets her over-optimism in the process of entrepreneurship. The entrepreneur's over-optimism hinders her real understanding of the entrepreneurial situation and damages her entrepreneurship competence. |
| | | I laugh all day and don't take anything to heart. A natural optimist. | optimism | |
| | | optimism is not necessarily a good thing. You see, I'm careless all day. I'm not as worried and melancholy as you, but my character is really bad for doing things. Impulsive and lack of consideration. | over optimism | |
| | | When I went to a plastic surgery hospital to do double eyelids (Common Asian plastic surgery. In recent years, Asian aesthetics has followed the European and American trend. Asian women yearn for double eyelids, long eyelashes, high-bridged noses, and white skin, so these aspects of plastic surgery are very popular—researcher's note), I saw that their business was so hot. I was so excited that I went to study for half a year and opened this store. I only saw the hot surface, not the hardships of this industry. | entrepreneurship competence (opportunity ability); over-optimism | |

**Table 4.** *Cont.*

| Material Code | Entrepreneur Code | Expression | Classification | Evaluation |
|---|---|---|---|---|
| | | When I finished double eyelid and false eyelashes for several friends around me, I found that there was no business to do. | entrepreneurship competence | |
| | | My mother said I was stupid and bold to open a shop. Take it for granted. I only saw the thief eat meat, but I didn't see the thief beaten. (Chinese proverbs. It means only seeing the good side of others, not the bad side—researcher's note) | entrepreneurship competence; over-optimism | |
| | | Fortunately, it was not a big investment. It should be regarded as the accumulation of experience of entrepreneurial failure. | optimism | |
| Material 16 | Entrepreneur 4 | I'm too stupid and naive. | over-optimism; entrepreneurship competence | The entrepreneur's over-optimism exacerbates his fluke psychology. In addition, his low education and legal blindness not only lead to entrepreneurial failure, but also bring about the negative situation of imprisonment. His entrepreneurial ability is "self-evident". |
| | | Then I was a little floating. I thought I can do anything. I also had a fluke mentality. Real jade and fake jade were sold together. At that time, I thought people always do in this way, and I wouldn't have an accident. As the saying goes, wealth is sought in danger. (Chinese proverb. It means you have to do something illegal to get rich. Obviously, the value orientation of this saying is incorrect—researcher's note) | over-optimism | |
| | | I went in (prison—researcher's note) for 11 months, and my jade store was sealed up. | entrepreneurship competence | |

*4.2. Quantitative Result*

4.2.1. Data Quality Analysis

Data quality analysis involves reliability and validity analyses, the homologous analysis of variance, descriptive analyses, and correlation analyses (Supplementary Materials: Data Quality Analysis).

Cronbach's $\alpha$ and the CITC value show that the reliability of the scale is generally ideal. The $X^2/df$ value, RMSEA value, IFI, TLI, CFI, PGFI, and PNFI show that the index adaptation of the scale is ideal. The CR and AVE indicate that the convergent validity of the scale meets the standard requirement. The Harman single-factor test indicates that common method variance does not exist in this research.

### 4.2.2. Hypothesis Test

Figure 2 and Table 5 suggest that the absolute fitting index and parsimony fitting index is ideal. Overall, the index adaptation of the mediation model is ideal.

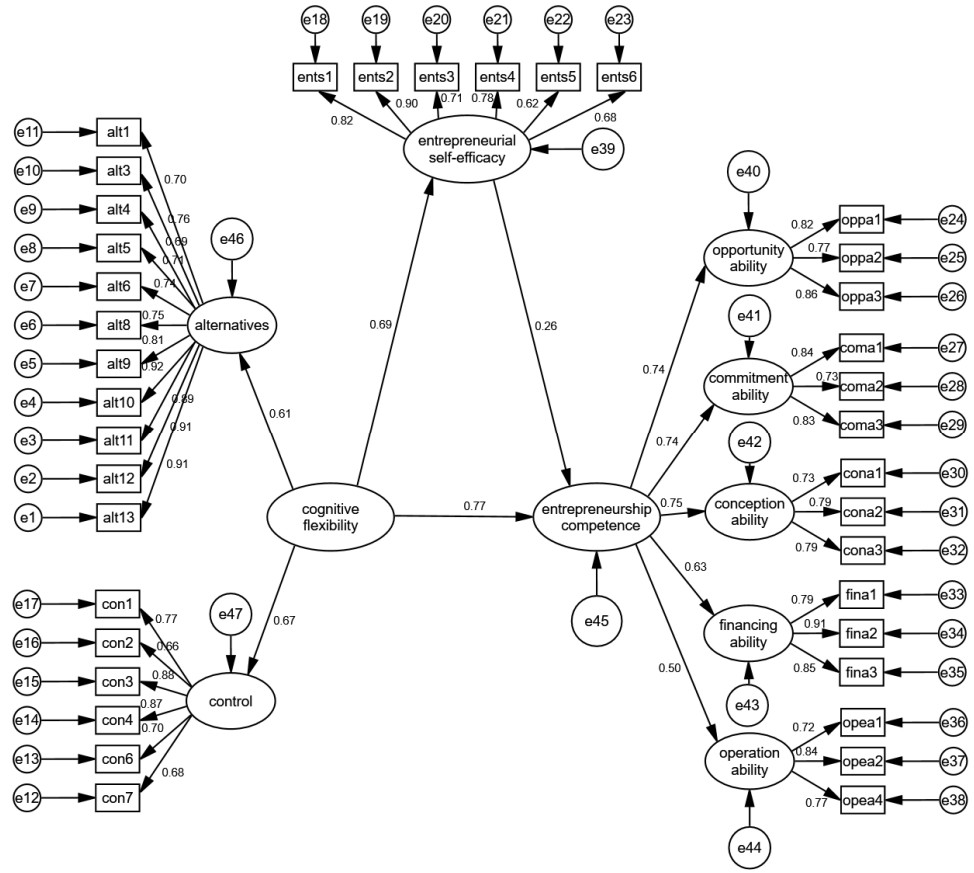

**Figure 2.** Mediation model.

**Table 5.** Mediation model fitting index.

| Index | Absolute Fit Index | | Incremental Fit Index | | | Parsimony Fit Index | |
|---|---|---|---|---|---|---|---|
| Specific classification | $X^2/\mathrm{df}$ | RMSEA | IFI | TLI | CFI | PGFI | PNFI |
| Judgment criteria | <5 | <0.08 | >0.9 | >0.9 | >0.9 | >0.5 | >0.5 |
| Fitness effect | 1.848 | 0.055 | 0.934 | 0.929 | 0.933 | 0.737 | 0.807 |

Table 6 shows that hypothesis 1 (H1), 2 (H2), and 3 (H3) have been confirmed.

**Table 6.** Path analysis.

| Path | | | Estimate | S.E. | C.R. | $p$ |
|---|---|---|---|---|---|---|
| entrepreneurial_self-efficacy | <— | cognitive_flexibility | 0.694 | 0.115 | 6.513 | *** |
| entrepreneurship_competence | <— | entrepreneurial_self-efficacy | 0.261 | 0.104 | 2.037 | 0.042 |
| entrepreneurship_competence | <— | cognitive_flexibility | 0.774 | 0.157 | 4.343 | *** |

*** means $p < 0.001$.

In order to verify the mediating effect more accurately, the bootstrap method is adopted. The bootstrap repeated sampling is performed five thousand times, and the confidence interval level is set to ninety-five percent. The sampling method adopts the nonparametric percentile method of deviation correction.

As Table 7 suggests, hypothesis 4 (H4) has been confirmed. As Table 8 suggests, hypothesis 5 (H5) has been confirmed.

**Table 7.** The mediating effect of entrepreneurial self-efficacy on the positive impact of cognitive flexibility on entrepreneurship competence.

| Effect | Standardized Estimation Coefficient | S.E. | LLCI | ULCI |
|---|---|---|---|---|
| Total effect | 0.955 | 0.079 | 0.800 | 1.115 |
| Direct effect | 0.744 | 0.271 | 0.509 | 1.386 |
| Indirect effect | 0.181 | 0.054 | 0.069 | 0.316 |

**Table 8.** Moderating role of optimism in the relationship between cognitive flexibility and entrepreneurship competence.

| | Unstandardized Coefficients | | Standard Coefficient | $t$ | Significance |
|---|---|---|---|---|---|
| | B | S.E. | BETA | | |
| (Constant) | 3.657 | 0.027 | | 136.701 | 0.000 |
| cognitive flexibility | 0.543 | 0.037 | 0.715 | 14.551 | 0.000 |
| optimism | 0.303 | 0.042 | 0.384 | 7.188 | 0.000 |
| optimism quadratic term | 0.078 | 0.028 | 0.167 | 2.840 | 0.005 |
| cognitive flexibility multiplying optimism | −0.172 | 0.062 | −0.159 | −2.750 | 0.006 |
| cognitive flexibility multiplying optimism quadratic term | −0.184 | 0.034 | −0.353 | −5.376 | 0.000 |

$R^2 = 0.487$ adj-$R^2 = 0.479$ F = 60.300 $p = 0.000$.

*Dependent variable: entrepreneurship competence.*

**5. Discussion: The Relation among Entrepreneurial Self-Efficacy, Cognitive Flexibility, and Open Innovation**

At the beginning, we mentioned that the positive effect of entrepreneurship competence on open innovation dynamics has been discussed in much research. Then, we empirically demonstrated the impact of entrepreneurial self-efficacy, cognitive flexibility, and other factors on entrepreneurship competence. Therefore, we assume that there are certain relations between entrepreneurial self-efficacy, cognitive flexibility, and open innovation.

In previous literature, the impact of entrepreneurial self-efficiency on open innovation has been mentioned. Individuals or organizations with a higher degree of entrepreneurial self-efficiency are more willing to take the initiative to innovate, and their willingness to innovate is stronger [62–64]; entrepreneurs' entrepreneurial self-efficiency level can also positively predict innovation ability [65–68]. Self-efficiency itself is the source and driving force of individual innovation [69,70].

Cognitive flexibility also has a positive impact on open innovation. Individuals with flexible cognition have an open rather than closed understanding of innovation; they not only take the internal resources of the enterprise as the driving force of innovation, but also integrate the external resources of the enterprise for innovation through flexible cognition and flexible problem handling [71–73]. For the overall open innovation strategy of enterprises, the cognitive flexibility of enterprise leaders and employees is an important non-material dynamic [74–77].

It can be seen that entrepreneurial self-efficiency and cognitive flexibility can not only directly affect the open innovation of enterprises, but also indirectly improve the open innovation level of enterprises by affecting the entrepreneurial ability of entrepreneurs. In addition to entrepreneurial ability, entrepreneurs' entrepreneurial self-efficacy and cognitive flexibility are also factors that enterprises and even society in general need to pay attention to.

# 6. Conclusions

## 6.1. Practical Implications

The interview result and questionnaire data suggest that, to improve the entrepreneurship competence of entrepreneurs, an effective starting point is to attach importance to the improvement of cognitive flexibility, entrepreneurial self-efficacy, and the effect of optimism.

On the one hand, from the perspective of entrepreneurs, firstly, it is suggested that they pay attention to the cultivation and improvement of their cognitive flexibility in the early stage of entrepreneurship, and in the processes of entrepreneurial practice. According to research results from Cambridge University, cognitive behavioral therapy and structured learning are effective means to improve cognitive flexibility [78]; both mindfulness mediation and non-invasive brain stimulation are good for cognition [79,80]. Secondly, self-efficacy is not a fixed personal characteristic. It can be developed through learning, education, and participation in relevant practical activities [81]. Entrepreneurs can enhance confidence in their entrepreneurial competence by strengthening their knowledge in the field of entrepreneurship and storing relevant entrepreneurial knowledge and information. Before starting a business, entrepreneurs should set reasonable entrepreneurial goals and should not aim too high, as unreachable goals will lead to low self-efficacy at the beginning. Thirdly, entrepreneurs are encouraged to emphasize the cultivation of positive emotions such as optimism. On the other hand, though, entrepreneurs may also be considering the entrepreneurial environment and reality, and be a rational optimist, so as to avoid the cognitive deviation caused by blind optimism.

On the other hand, from the perspective of the general environment and atmosphere, society should realize that some cognitive and psychological elements in the entrepreneurial process are not limited to the personal perceptions of individual entrepreneurs but are also important entrepreneurial resources and dynamics of open innovation [82]. It should be noted that entrepreneurs are not isolated individuals; they exist within certain social relations. They are also someone's parent, child, husband, wife, friend, and/or colleague. Relevant research shows that positive emotions can promote the development of individual cognitive flexibility [38]. In light of this, it is essential for an entrepreneur's social relations to actively care about their psychological situation, understand them, give them warm care and support, and create a comfortable atmosphere more suitable for the improvement of positive emotions, as well as entrepreneurial ability.

Moreover, in the context of COVID-19, entrepreneurs are generally depressed [20]. The government should create a more friendly entrepreneurial environment and give entrepreneurs more policy support and spiritual encouragement, so as to make them more confident in their entrepreneurship and relatively optimistic about the entrepreneurial situation, and thus continuously stimulate their positive psychological conditions and promote the improvement of their entrepreneurship competence as well as the open innovative atmosphere at the whole social level [83,84].

The literature shows that, in addition to the cognitive and psychological factors discussed in this study, entrepreneurship competence is also affected by many other factors; for example, the entrepreneur's gender, age, educational level, personality, etc., as well as external factors such as company size and policies [85]. This reminds us that the improvement of entrepreneurship competence is not limited to one aspect. The enhancement of entrepreneurship competence requires holistic effort. Because of the many factors involved, the optimization of entrepreneurship competence is not achieved overnight. This requires entrepreneurs to have patience and confidence. Therefore, focus has returned to the psychological, cognitive level. It is very important to have a long-term vision and a resilient attitude.

During the interviews, we found an interesting phenomenon: the attributes of entrepreneurs showing competence seem to reflect the characteristics of the times. We noticed that, in the early period—entrepreneurship in the period of the reform and opening up of China—superb entrepreneurship competence is usually accompanied by a fearless attitude

of daring to venture. Generally speaking, courage is a significant part of, or an influencing factor in, entrepreneurship competence and spirit. For entrepreneurs beginning in the 1980s and 1990s, and other younger generations, higher entrepreneurship competence is usually related to prudence. Compared with the passionate tone shared by the capable entrepreneurs of previous generations, the new generations of entrepreneurs with positive preliminary achievements mostly show a cautious temperament; their self-efficacy and optimism are accompanied by careful measurement and thinking. This reminds us that entrepreneurship competence is related to the characteristics of the times and requires entrepreneurs to advance with the times and lay emphasis on the requirements of the social moment.

*6.2. Research Limitations and Future Prospects*

This research may have certain limitations as regards the research perspective and operation methods, which are expected to be improved in future research.

Firstly, this research was carried out at the individual level. In fact, studies on entrepreneurship competence performed at the organizational level may have different levels of practical significance, such as for companies, industries, and governments. In light of this, studies on the improvement of entrepreneurship at the organizational level can be carried out in the future. Secondly, the sample is not comprehensive. For one thing, due to limited resources, this research could not carry out random sampling in the strict sense and could only rely on the non-probability sampling method, which leads to a reduction in sample comprehensiveness to some extent. At the same time, due to the limitations of policy during the epidemic, this study did not conduct more extensive and comprehensive interviews; otherwise, more voices may have been heard, and more enlightening interview results may have been obtained. In the future, researchers can conduct more large-scale questionnaires and more in-depth interviews to obtain more objective research conclusions. Thirdly, one of the tools used in this research is the five-point scale, which defines entrepreneurs' cognition, effectiveness, and optimism according to their choices of items. This approach lacks strong objectivity. Due to the limitations of time as well as our shoestring budget, this research could not use the experimental and measurement tools often employed in psychology, such as the reaction time meter, the Yekes selector, EEG, and nuclear magnetic resonance technology, which can more objectively monitor the psychological characteristics of entrepreneurs. Future entrepreneurship research is expected to learn more from other psychological methods and tools.

**Supplementary Materials:** The following supporting information can be downloaded at: https://www.mdpi.com/article/10.3390/joitmc8020065/s1, Basic Information of Interviewees; General Situations of the Sample; Collection of Interview Materials; Data Quality Analysis.

**Author Contributions:** Conceptualization, W.Y. and J.A.; methodology, W.Y. and J.A.; software, W.Y.; validation, W.Y. and J.A.; formal analysis, W.Y.; investigation, W.Y.; resources, W.Y.; data curation, W.Y.; writing—original draft preparation, W.Y.; writing—review and editing, J.A.; visualization, W.Y.; supervision, J.A. All authors have read and agreed to the published version of the manuscript.

**Funding:** This research received no external funding.

**Institutional Review Board Statement:** Not applicable.

**Informed Consent Statement:** Not applicable.

**Data Availability Statement:** Not applicable.

**Acknowledgments:** Thanks to Fujian Normal University and University of Liverpool.

**Conflicts of Interest:** The authors declare no conflict of interest.

**Appendix A**

Cognitive Flexibility, Entrepreneurial Self-efficacy, Optimism,
and Entrepreneurship Competence Questionnaire

Dear entrepreneurs,

We are conducting a research on entrepreneurs and hereby issue this questionnaire. The questionnaire is anonymous and does not disclose privacy. The questionnaire options vary from person to person and do not involve pros and cons, right and wrong. If you can tick the options that reflect your real situation, it will be very beneficial to our research.

Pay tribute to entrepreneurs.

Part I. Cognitive Flexibility

The following are the measurement items of cognitive flexibility.

Options 1 to 5 gradually increase the degree of compliance.

Please tick the option that best suits your situation.

1. I am good at evaluating and analyzing all kinds of situations and states. 1□ 2□ 3□ 4□ 5□

2. Before making a decision, I will consider different options. 1□ 2□ 3□ 4□ 5□

3. I always look at difficulties from different perspectives. 1□ 2□ 3□ 4□ 5□

4. When analyzing the cause, I will search for additional information to help judge. 1□ 2□ 3□ 4□ 5□

5. When analyzing problems, I will try to consider the views of others. 1□ 2□ 3□ 4□ 5□

6. I'm good at seeing things from other people's perspectives. 1□ 2□ 3□ 4□ 5□

7. It is important to look at the dilemma from multiple perspectives. 1□ 2□ 3□ 4□ 5□

8. When deciding how to deal with difficulties, I will weigh different solutions. 1□ 2□ 3□ 4□ 5□

9. When analyzing problems, I often adopt new ideas. 1□ 2□ 3□ 4□ 5□

10. When explaining why something happens, I will consider all the facts and information. 1□ 2□ 3□ 4□ 5□

11. In the face of difficulties, I will calm down and think about a variety of solutions. 1□ 2□ 3□ 4□ 5□

12. I can think of more than one way to solve the difficulties I face. 1□ 2□ 3□ 4□ 5□

13. Before dealing with a problem, I will consider a variety of ways to deal with it. 1□ 2□ 3□ 4□ 5□

14. I don't know how to make a decision in the face of a difficult situation. 1□ 2□ 3□ 4□ 5□

15. When I am in trouble, I feel powerless to control the situation. 1□ 2□ 3□ 4□ 5□

16. I feel so anxious in the face of difficulties that I can't think of any solution. 1□ 2□ 3□ 4□ 5□

17. When there are multiple solutions to a problem, I feel at a loss. 1□ 2□ 3□ 4□ 5□

18. I'm at a loss when I'm in trouble. 1□ 2□ 3□ 4□ 5□

19. I have the ability to solve problems in life. 1□ 2□ 3□ 4□ 5□

20. In the face of difficulties, I feel I can't change anything. 1□ 2□ 3□ 4□ 5□

Part II. Entrepreneurial Self-efficacy

The following are the measurement items of entrepreneurial self-efficacy.

Options 1 to 5 gradually increase the degree of compliance.

Please tick the option that best suits your situation.

21. It is easy for me to start a business and keep it running. 1□ 2□ 3□ 4□ 5□

22. I am ready to create a viable company. 1□ 2□ 3□ 4□ 5□

23. I have the ability to control the process of creating a new business. 1□ 2□ 3□ 4□ 5□

24. I understand the practical details required to start a business. 1□ 2□ 3□ 4□ 5□

25. I know the way to produce an entrepreneurial program. 1□ 2□ 3□ 4□ 5□

26. If I start a company, I have a good chance of success. 1□ 2□ 3□ 4□ 5□

Part III. Optimism

The following are the measurement items of optimism.

Options 1 to 5 gradually increase the degree of compliance.

Please tick the option that best suits your situation.

27. I often think things for the best when I'm not sure. 1☐ 2☐ 3☐ 4☐ 5☐

28. If something has the possibility of going bad, it will really go wrong. 1☐ 2☐ 3☐ 4☐ 5☐

29. I always think the future is bright. 1☐ 2☐ 3☐ 4☐ 5☐

30. I never expect to get what I want. 1☐ 2☐ 3☐ 4☐ 5☐

31. I seldom expect good things to happen to me. 1☐ 2☐ 3☐ 4☐ 5☐

32. Generally speaking, I believe there will be more good things than bad things. 1☐ 2☐ 3☐ 4☐ 5☐

Part IV. Entrepreneurship Competence

The following are the measurement items of entrepreneurship competence.

Options 1 to 5 gradually increase the degree of compliance.

Please tick the option that best suits your situation.

33. I can identify potential market areas. 1☐ 2☐ 3☐ 4☐ 5☐

34. I can assess the strengths and weaknesses of potential business opportunities. 1☐ 2☐ 3☐ 4☐ 5☐

35. I can seize high-quality business opportunities and implement them. 1☐ 2☐ 3☐ 4☐ 5☐

36. I can endure all kinds of pressure and unexpected changes in my work. 1☐ 2☐ 3☐ 4☐ 5☐

37. I will stick to my career even in the face of adversity. 1☐ 2☐ 3☐ 4☐ 5☐

38. I will keep my promise and be fair, open-minded, and honest in market activities and enterprise management. 1☐ 2☐ 3☐ 4☐ 5☐

39. I can link relevant ideas, questions, and observations from different resources. 1☐ 2☐ 3☐ 4☐ 5☐

40. I will timely adjust the company's strategic objectives and business ideas. 1☐ 2☐ 3☐ 4☐ 5☐

41. I can accurately reposition the position of the enterprise in the market. 1☐ 2☐ 3☐ 4☐ 5☐

42. I can develop effective ways to finance. 1☐ 2☐ 3☐ 4☐ 5☐

43. I can use various ways to finance. 1☐ 2☐ 3☐ 4☐ 5☐

44. I can successfully obtain the government's policy financial support. 1☐ 2☐ 3☐ 4☐ 5☐

45. I can effectively lead, supervise, and motivate employees. 1☐ 2☐ 3☐ 4☐ 5☐

46. I can reasonably allocate human, financial, material, and other resources within the enterprise. 1☐ 2☐ 3☐ 4☐ 5☐

47. I can build and maintain relationships with people with key resources. 1☐ 2☐ 3☐ 4☐ 5☐

48. I can take remedial measures in time to solve the problems and difficulties in the company's operation. 1☐ 2☐ 3☐ 4☐ 5☐

Part V. Basic Information

49. Your gender

A. male

B. female

50. Your age

A. under 35

B. 35–50

C. over 50

51. Your education level
A. below junior college
B. junior college
C. undergraduate
D. postgraduate or above
52. Your industry
A. agriculture, forestry, animal husbandry, fishery
B. mining, manufacturing, power, heat, gas, and water production and supply, construction
C. transportation, storage and postal services, information transmission, computer services and software, wholesale and retail, accommodation and catering, finance, real estate, leasing and business services, scientific research, technical services and geological exploration, water conservancy, environment and public facilities management, residents' services and other services, education, health, social security and social welfare, culture, sports and entertainment, public management and social organizations, international organizations and other industries
53. The scale of your business
A. extra-large, large
B. medium
C. small
D. miniature
Thank you very much for your support.

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
