# Peer review of "Cognitive Mechanisms in Entrepreneurship Competence: Its Implication for Open Innovation"

_2199-8531, doi:10.3390/joitmc8020065_

Round 1

Reviewer 1 Report

Author/-s should improve/rethink:

  • in Abstract - main aim of the article should be added
  • in Introduction - the novelty of the research should be better pointed; the structure of the paper shpuld be presented at the end of this part.
  • in Methods - the scope of the research should absolutely be added - based on the discussion, it can be assumed that the research was performed in China, but it should be clearly specified in the Method - not only the country, but also the region should be indicated.
  • In Discussion - practical implications of the research should be described

Author Response

Dear Sir or Madam,

Thank you for your suggestions for our article. All the suggestions are quite pertinent.

We revised the article according to your suggestion. Specifically:

in Abstract - main aim of the article should be added

Completed, please refer to the attachment.

in Introduction - the novelty of the research should be better pointed; the structure of the paper should be presented at the end of this part.

Completed, please refer to the attachment. According to opinion of editors, literature review is needed, so we may add it later, and so we mentioned literature review in the structure.

in Methods - the scope of the research should absolutely be added - based on the discussion, it can be assumed that the research was performed in China, but it should be clearly specified in the Method - not only the country, but also the region should be indicated.

Completed, please refer to the attachment.

In Discussion - practical implications of the research should be described

Completed. We further subdivided the Discussion into practical enlightenment and research limitations. Please refer to the attachment.

Thank you again for your help.

All the best.

Wenjing and Jose.

Reviewer 2 Report

The paper deals with an important topic but the mixed method approach is a bit unfocused, with a lot of information but not a clear narrative in terms of contribution and implications.

The first phrase of the paper is important to set up a tone and I think that the existing one is rather unclear. Entrepreneurship as a disposition? Maybe entrepreneurial orientation. Then, it’s rather strong to say that it aims to create economic growth – entrepreneurs aim to create value, and the sector contributes to economic growth.

In a similar vein, I would suggest replacing attributes like “huge” and “breathtaking” (lines 25 and 26) with more informative and less bombastic synonyms (the same thing in lines 33 and 96 “undoubtedly” – there are very few findings without doubt in the social and behavioral sciences).

When claiming the need to improve entrepreneurial competences, a previous link must be set between lack of competences and entrepreneurial failure (with proper references).

Line 79 can be easily eliminated.

When presenting hypotheses, the authors need to start each subsection with a clear definition of the concepts involved. And only then to present supporting evidence. For instance, when discussing optimism, it is important to clarify if you are talking about dispositional optimism or rather about an optimism bias (I see it from the scale that it is the first instance but it has to be clear in the text also).

I do see the role of working with mixed methods but given the type of hypotheses that the authors propose, the expectation is on testing them with the quantitative approach. It’s actually hard to understand what is the exact contribution of the qualitative part – there are two types of explorations, what are the distinctive findings? (they converge, they don’t? if not, to what extent the sample for the qualitative interviews may be biased?).

 Anyways, you should avoid expressions/subsections like “the effect of optimism” when simply looking at interviews (you cannot identify effects, maybe just associations).

When reporting quantitative findings, it’s important not to repeat the information from tables/figures into the main text. Also, do not repeat hypotheses, you’ve numbered them and it is enough to refer to them as H1, H2 etc.

You say: As shown in Table 8, cognitive flexibility is the independent variable, optimism is the 405 moderating variable, and entrepreneurship competence is the dependent variable. (lines 405). This sounds like you’ve did an exploration to see what is the dependent and what is the independent. You may want to remove this.

The findings do not mention the control variables: are there any differences by gender, age, education, industry, business scale?

Author Response

Dear Sir or Madam,

Thank you for your suggestions for our article. All the suggestions are quite pertinent (some opinions have moved one of the authors inexplicably).

We revised the article according to your suggestion. Specifically:

The paper deals with an important topic but the mixed method approach is a bit unfocused, with a lot of information but not a clear narrative in terms of contribution and implications.

As for the starting point of adopting the hybrid method, we made a supplementary explanation in the innovation of the article; at the same time, we also explained some unexpected gains of qualitative interview in practical enlightenment. Please refer to the blue font of the attachment for the above. Unfortunately, this study is one of the authors’ first exploration of mixed methods, so it seems childish in the coordination of methods and the expression of results. She will continue to study and work hard in the future.

The first phrase of the paper is important to set up a tone and I think that the existing one is rather unclear. Entrepreneurship as a disposition? Maybe entrepreneurial orientation. Then, it’s rather strong to say that it aims to create economic growth – entrepreneurs aim to create value, and the sector contributes to economic growth.

Completed, please refer to the attachment.

In a similar vein, I would suggest replacing attributes like “huge” and “breathtaking” (lines 25 and 26) with more informative and less bombastic synonyms (the same thing in lines 33 and 96 “undoubtedly” – there are very few findings without doubt in the social and behavioral sciences).

Completed, please refer to the attachment. Your suggestion is very enlightening: in future writing, we will also try to be rigorous and try to avoid arbitrary expression.

When claiming the need to improve entrepreneurial competences, a previous link must be set between lack of competences and entrepreneurial failure (with proper references).

Completed, please refer to the attachment.

Line 79 can be easily eliminated.

Completed, please refer to the attachment. You are very careful.

When presenting hypotheses, the authors need to start each subsection with a clear definition of the concepts involved. And only then to present supporting evidence. For instance, when discussing optimism, it is important to clarify if you are talking about dispositional optimism or rather about an optimism bias (I see it from the scale that it is the first instance but it has to be clear in the text also).

Completed, please refer to the attachment.

I do see the role of working with mixed methods but given the type of hypotheses that the authors propose, the expectation is on testing them with the quantitative approach. It’s actually hard to understand what is the exact contribution of the qualitative part – there are two types of explorations, what are the distinctive findings? (they converge, they don’t? if not, to what extent the sample for the qualitative interviews may be biased?).

In fact, the hypotheses in this article are only suitable for quantitative research. The main audience of qualitative research is actually entrepreneurs. Therefore, a better way for similar research in the future is to write another article about the qualitative research part. Thank you for your reminder.

Anyways, you should avoid expressions/subsections like “the effect of optimism” when simply looking at interviews (you cannot identify effects, maybe just associations).

Changed, please refer to the attachment.

When reporting quantitative findings, it’s important not to repeat the information from tables/figures into the main text. Also, do not repeat hypotheses, you’ve numbered them and it is enough to refer to them as H1, H2 etc.

Completed, please refer to the attachment. This suggestion is very useful for our future writing, and we are very moved.

You say: As shown in Table 8, cognitive flexibility is the independent variable, optimism is the 405 moderating variable, and entrepreneurship competence is the dependent variable. (lines 405). This sounds like you’ve did an exploration to see what is the dependent and what is the independent. You may want to remove this.

Completed, please refer to the attachment.

The findings do not mention the control variables: are there any differences by gender, age, education, industry, business scale?

We analyzed the above control variables in the SUPPLEMENTARY MATERIALS. But they were not examined in this study. Literature shows that many factors will affect a person’s ability, such as socio-economic environment and some demographic capital variables, such as gender, age, family background, and education level. This study does not intend to involve objective factors or factors that are relatively difficult to change that affecting entrepreneurship competence, as these factors can easily be reversed by non entrepreneurs themselves, but only from some individual and subjective factors that entrepreneurs can adjust or operate through efforts. However, in future research, we will pay attention to the influence of control variables.

We will further optimize the language and details before submitting the final version to the editor.

Thank you again for your help.

All the best.

Wenjing and Jose.

Reviewer 3 Report

Dear Authors,

I have mixed feelings about your manuscript. The topic is interesting but there are many issues, both in terms of conceptualization and treatment of data.

Introduction

  • “Entrepreneurship is a disposition [..]”. This statement needs to be referenced.
  • The very core of the manuscript, entrepreneurship competence, is not defined. Moreover, I do not understand why the authors use entrepreneurship competence and not entrepreneurial competence. In terms of competencies, OECD addressed them (Developing entrepreneurship competencies, Mexico City, 2018). Therefore, clarification is needed.
  • What are the differences between antecedent variables and influencing elements of entrepreneurship competence? What were the criteria to differentiate between them?
  • I do not see the point in highlighting COVID-19, it is marginal in the context of the manuscript.

Research Hypotheses

  • A research hypothesis cannot be stated as “is likely to…”.
  • The conceptual model is not clearly explained. For instance, why those antecedents were chosen by the authors.

Method and Results

  • I do not see the actual equation(s) describing the model. What were the covariates?
  • This statement puzzles me: “The main criteria for excluding questionnaires are: [..] the attitude is obviously perfunctory”. How was the attitude measured to enable the authors to reject questionnaires based on this criterion?
  • It seems the authors had used SEM. What software was used for the analysis? Overall, very little information is provided.
  • I do not see the point in investigating various people attending a conference instead of real entrepreneurs.
  • Regarding H5, I do not see how the authors used SEM to assess an inverted U-shaped moderation effect.
  • Overall, the authors need to present in detail the constructs if these were reflective or not. Common method bias is not addressed at all.

Author Response

Dear Sir or Madam,

Thank you for your suggestions for our article. Some of the suggestions are quite pertinent.

According to your suggestion, we rearranged our ideas and moderately revised the article.

Specifically:

I have mixed feelings about your manuscript. The topic is interesting but there are many issues, both in terms of conceptualization and treatment of data.

One of the authors is also a journal editor. She deeply understands your mood. Thank you for your valuable time for our article.

Introduction

“Entrepreneurship is a disposition [..]”. This statement needs to be referenced.

The very core of the manuscript, entrepreneurship competence, is not defined. Moreover, I do not understand why the authors use entrepreneurship competence and not entrepreneurial competence. In terms of competencies, OECD addressed them (Developing entrepreneurship competencies, Mexico City, 2018). Therefore, clarification is needed.

We have changed some inappropriate expressions. The definition of key concepts is added in part 2. Please refer to the attachment, blue font.

“Entrepreneurship competence” is the translation of an authoritative magazine in China. We adopted it. But we think “entrepreneurial competence” is more accurate. In future research, we will adopt your suggestions.

What are the differences between antecedent variables and influencing elements of entrepreneurship competence? What were the criteria to differentiate between them?

Compared with the “influencing elements”, the “antecedent variables” focus more on the “generation” mechanism, which is more fundamental. In this study, we did not distinguish between the two concepts.

I do not see the point in highlighting COVID-19, it is marginal in the context of the manuscript.

Affected by the epidemic, many people have even less confidence in entrepreneurship. Please refer to the attachment for specific explanation, green font.

Research Hypotheses

A research hypothesis cannot be stated as “is likely to…”.

Changed. Please refer to the attachment, red font.

The conceptual model is not clearly explained. For instance, why those antecedents were chosen by the authors.

The introduction mentioned that entrepreneurial cases and theories show the relationship between these antecedents and entrepreneurial competence. Literature showed that many factors will affect a person’s ability, such as socio-economic environment and some demographic capital variables, such as gender, age, family background, and education level. This study does not intend to involve objective factors or factors that are relatively difficult to change that affecting entrepreneurial competence, as these factors can easily be reversed by non entrepreneurs themselves, but only from some individual and subjective factors that entrepreneurs can adjust or operate through efforts.

Method and Results

I do not see the actual equation(s) describing the model. What were the covariates?

Due to space constraints, the analysis of sample conditions, the test of data quality, covariates and other contents that are not particularly close to the research assumptions are in the SUPPLEMENTARY MATERIALS.

This statement puzzles me: “The main criteria for excluding questionnaires are: [..] the attitude is obviously perfunctory”. How was the attitude measured to enable the authors to reject questionnaires based on this criterion?

Some questionnaires choose the same option from beginning to end, such as 5. We judge that the respondents only want to finish the answer quickly and have a perfunctory attitude.

It has been explained. Please refer to the red font in the corresponding part of the attachment.

It seems the authors had used SEM. What software was used for the analysis? Overall, very little information is provided.

Amos. Please refer to the attachment, yellow font.

I do not see the point in investigating various people attending a conference instead of real entrepreneurs.

They are entrepreneurs who attend an entrepreneur conference. Please refer to the attachment, purple font.

Regarding H5, I do not see how the authors used SEM to assess an inverted U-shaped moderation effect.

Please refer to Table 8. Among them, cognitive flexibility is independent variable, optimism is moderating variable, and entrepreneurship competence is dependent variable. The test results suggest that the influence coefficient of the interaction term between cognitive flexibility and optimism is -0.172, and it is significant. The interaction coefficient of cognitive flexibility and optimism quadratic term is significant, and the regression coefficient is -0.184, which is significantly negative.

Overall, the authors need to present in detail the constructs if these were reflective or not. Common method bias is not addressed at all.

We analyzed the above in the SUPPLEMENTARY MATERIALS.

Thank you again for your help.

Later, we will optimize the language and details of the article according to the suggestions of yours and the editor's.

All the best.

Alexandra Yang and José Alves.

Round 2

Reviewer 2 Report

Thank you for the revised and improved version.

Some final suggestions for refining the manuscript:

  • Streamline the abstract. It is too long. It could easily start from line 54. The idea that qualitative research has revealed correlations is too general to be of interest (and I would avoid correlations since the term is rather quantitative). Instead of questionnaire date you may want to mention that the SEM analysis reveals different findings.
  • Consider removing completely figure 1 and eventually replacing it with a paragraph of the type: the rest of the paper is organized as follows: section 2, section 3 etc.
  • Line 199: consider replacing “the innovation of this research” with “the contribution of this research”
  • Please also introduce the sign of the effect in figure 2: e.g. H1(+)
  • Consider replacing “ a hypothesis tends to be reliable” with “the hypothesis has been confirmed”.
  • Lines 961-962 can be eliminated. Both the literature review and the interviews are also part of your research paper.
  • Shorten some of the longer phrases to increase readability. For instance lines 164-168, 222-226, 940-947

Author Response

Dear Sir or Madam,

Thank you for taking the trouble to put forward valuable modification suggestions for our article. In this process, in addition to excellent professional standards, you also showed great patience and meticulous.

We have further improved the article according to your suggestions. Please refer to the attachment.

Best.

Wenjing and Jose.

P.S.

Anonymous review makes us unable to know your information. We look forward to keeping in touch with you in the future. Our email address is shuttlediplomacy@163.com.

Many thanks.

Reviewer 3 Report

The authors,

You had answered my comments and suggestions.

Author Response

Dear Sir or Madam,
Thank you for your approval.
The latest revision is for your reference.
We wish you all the best,
Alexandra and Jose.
